# Compensation of Temperature-Induced Errors in Quartz Flexible Accelerometers Using a Polynomial-Based Non-Uniform Mutation Genetic Algorithm Framework

**DOI:** 10.3390/s25030653

**Published:** 2025-01-23

**Authors:** Jinyue Zhao, Kunpeng He, Kang Le, Yongqiang Tu

**Affiliations:** 1College of Artificial Intelligence, Nankai University, Tianjin 300350, China; 2College of Marine Equipment and Mechanical Engineering, Jimei University, Xiamen 361021, China; 202361000149@jmu.edu.cn

**Keywords:** quartz flexible accelerometer (QFA), temperature error modeling, parameter identification, error compensation, strapdown inertial navigation system (SINS)

## Abstract

The quartz flexible accelerometer (QFA) is a critical component in navigation-grade strapdown inertial navigation systems (SINS) due to its bias error, which significantly impacts the overall navigation accuracy of SINS. Temperature variations induce dynamic changes in the bias and scale factor of QFA, leading to a degradation of the navigation accuracy of SINS. To address this issue, this paper proposes a temperature error compensation method based on a non-uniform mutation strategy genetic algorithm (NUMGA) and a polynomial curve model (PCF). Firstly, the temperature bias mechanism of QFA output is analyzed, and a polynomial temperature error model is established. Then, the NUMGA is utilized to identify the model parameters using the −20–40 °C test data, seeking the optimal parameters for the polynomial. Finally, the compensation parameters are used for cold start static test verification. The results demonstrate that the temperature compensation model based on NUMGA-PCF can automatically select the optimal parameters, which enable the model to exhibit a stable decreasing trend on the adaptation curve without multiple fluctuations. Compared to the traditional GA temperature compensation model, the compensation errors in the three axes of QFA in SINS are reduced by 612.24 μg, 60.82 μg, and 875.82 μg, respectively. Before the 20th generation, there are no decrease in convergence speed observed with the in-crease of population diversity. Within the −20–40 °C temperature range, the average values and standard deviations of QFA for the three optimized axes can be maintained below 0.1 μg by using this compensation model.

## 1. Introduction

The strapdown inertial navigation system (SINS) is composed of three gyroscopes and three accelerometers mounted orthogonally. And SINS can provide real-time navigation information including position, velocity, and attitude. SINS has been widely used in civilian applications like unmanned vehicles, oil and gas exploration, consumer electronics, as well as military fields including aviation, navigation, and high-end weapon systems [1,2,3]. Gyroscope and accelerometer are essential sensor components of SINS, responsible for measuring angular velocity and acceleration, respectively [4,5,6]. However, the navigation accuracy of SINS highly depends on environmental factors such as temperature, geomagnetic fields, vibrations and shocks [7,8]. Temperature sensitivity is a common issue in SINS, which is characterized by the instability of bias and scale factor of accelerometers and gyroscopes caused by temperature variations. These changes can introduce non-linear characteristics and temperature gradient effects in sensor outputs, thereby the accuracy and stability of SINS are impacted [9,10]. Because the navigation and positioning principle of SINS is based on dead reckoning, the navigation error increases caused by inertial device errors with time. And the bias errors of gyroscopes and acceleration will lead to large navigation error for SINS [11].

Gyroscope type in SINS includes mechanical gyro, fiber optic gyro, laser gyro, atomic gyro and micro electro mechanical systems (MEMS) gyro [12,13]. Accelerometer type in SINS includes pendulum integral gyro accelerometer (PIGA), QFA, resonant beam accelerometer and MEMS accelerometer [14]. Laser gyro and fiber optic gyro are typically used for navigation-grade SINS and QFA is widely used for acceleration measurements [15]. Compared with fiber optic gyro and laser gyro, the bias of QFA is greatly affected by temperature, which seriously limits the navigation accuracy of SINS [16].

The modeling and compensation of QFA temperature errors are extensively studied by researchers to address their effects on navigation-grade SINS. The frequency response characteristics of QFA under different temperature conditions are analyzed by Wang et al. [17]. It is found that in high-temperature environments, the frequency response exhibits significant instability. The temperature error effect can be also effectively suppressed by hardware solutions such as thermal isolation and heat dissipation settings, and the inclusion of temperature control devices [18,19]. For example, the relative stability of the equipment temperature in high or low-temperature environments is maintained through the design of new heat dissipation materials. These materials are adjusted for their thermal conductivity and adaptability. Through the rational arrangement of heat dissipation structures, the negative impact of temperature variations on the performance of navigation devices is mitigated. However, the temperature lag effect of accelerometers in SINS has been seldom considered [20]. The mechanical structure of QFA can be enhanced by using new materials with a low coefficient of thermal expansion, which restricts their temperature range. Consideration needs to be given to various aspects such as mechanical strength, hardness, and cost to ensure sufficient mechanical performance [21,22,23,24]. When the data transmission is not enough, the feedback signal of the temperature will be delayed, which will lead to the attitude solution error of the system [25,26,27]. Therefore, besides hardware solutions, software compensation is required to further suppress temperature bias errors caused by temperature variations in QFA.

Although the complexity of program is increased, there is no need to modify the hardware structure of the QFA for software compensation. Zhou et al. [28]. proposed an adaptive calibration algorithm based on temperature sensors, effectively reducing the impact of temperature on the measurement results. Furthermore, in-depth analyses were conducted by researchers on the temperature influence of accelerometer output error. Various effective algorithm compensation methods were proposed [29]. These methods include the least squares method (LSM), Kalman filters, and artificial neural networks [30]. The relationship between temperature and accelerometer output was optimized by LSM through mathematical optimization, effectively compensating for output error. Recursive filtering was employed by Kalman filters to estimate and update the system state, dynamically adjusting for error induced by temperature [31]. Complex nonlinear mapping relationships were established by artificial neural networks through learning from historical data, achieving more precise temperature compensation effects. These compensation methods were extensively validated in practical applications, significantly improving the performance of accelerometers in different scenarios [32,33].

However, there are some shortcomings of existing software compensation methods such as model dependence, high computational complexity, overfitting, and poor interpretation. To overcome these issues, a polynomial model was used in this study to establish the relationship between temperature error and QFA output. A non-uniform mutation genetic algorithm was employed to optimize the polynomial curve model (NUMGA-PCF) for modeling and compensating temperature error. The effectiveness of this software approach was validated through experimental data and performance evaluation. The algorithm increases the exploration ability of solution space by introducing nonlinear variation of variation rate. Compared to algorithms with fixed mutation rates, the proposed method focuses on extensive exploration in the early stage. In the later stages, greater emphasis is placed on local search to improve the algorithm convergence and search efficiency to find the optimal coefficients for the polynomial. The high adjustability of the algorithm enables it to flexibly meet the requirements of different scenarios. Compared to traditional software compensation methods, higher accuracy and stability are exhibited by NUMGA in practical applications. It provides a feasible and effective solution for the accelerometer temperature error compensation.

The structure of this article is as follows. Section 2 provides an analysis of the temperature error in Quartz Flexible Accelerometers (QFAs). In Section 3, the modeling and compensation methods for temperature error in QFAs are detailed. Section 4 describes the experimental setup used for validation, while Section 5 presents the results and discussion based on the experimental findings. Finally, Section 6 concludes the study.

## 2. Analysis of Temperature Error of QFA

The QFA is composed of an elastic structure, quartz crystal, and sensor circuit. The elastic structure, which is usually made of flexible materials, supports and secures the quartz crystal. Changes in temperature alter the dimensions and elastic modulus of the quartz crystal, thereby affecting the output signal of the QFA. The working principle of the QFA is illustrated in Figure 1. Under external acceleration, the inertial mass experiences slight displacements. This alters the spacing between the plates of the differential capacitor, resulting in a variation in differential capacitance. Subsequently, a servo amplifier detects these changes and generates a corresponding feedback current, which is transmitted to a coil located in the magnetic field generated by the magnets. The feedback moment generated in the coil counterbalances the input acceleration moment, causing the inertial mass to return to its equilibrium position. The self-heating of the coil introduces output errors that can be mathematically represented as:(1)$$\delta =(1+\beta_{1}\Delta T)(1+\beta_{2}\Delta T)-1$$
where *β*_1_ and *β*_2_ are temperature coefficients related to the coil and magnet, respectively, and Δ*T* represents the temperature change. *δ* corresponds to the output error introduced due to the self-heating effect.

The relationship between the feedback current and the input acceleration is as follows:(2)$$i=K_{0}+\frac{M}{BL}a$$
where *i* represents the feedback current, and *K*_0_ is the bias of the QFA. *M* denotes the mass of the quartz pendulum assembly, *B* is the magnetic flux density, and *L* refers to the length of the torque coil. The input acceleration is represented by *a*. This formula captures the interplay between the mechanical and electrical properties of the QFA, ensuring that the feedback current compensates for the input acceleration effectively and returns the inertial mass to its equilibrium position.

The high sensitivity and reliability of QFA to external accelerations are ensured by the implementation of a closed-loop feedback system. The major source of error in QFA is temperature bias caused by temperature changes. Changes in bias and scale factor of QFA are induced by temperature bias. Taking temperature bias, temperature sensitivity, and temperature variation into account, the temperature characteristics of QFA can be described by (3):(3)$$E=(K_{0}+P\cdot T_{0})+(S_{a}+P)\cdot T+\sum_{n=1}^{N}C_{n}\cdot {\left(T-T_{0}\right)}^{n}$$
where *E* represents the output of QFA. *K*_0_ denotes the bias. *T* represents the current temperature. *T*_0_ is the reference temperature. *P* is the temperature bias coefficient, which indicates the change in output caused by a unit temperature variation; the common unit is ±ppm/°C. *S*_a_ represents the temperature sensitivity. *C_n_* represents the coefficients corresponding to the terms in the *N*-th order expansion.

## 3. Modeling and Compensation of Temperature Error of QFA

### 3.1. Modeling of Temperature Error of QFA

Due to the temperature characteristics during cold starts, a gradual increase in the internal temperature of SINS environment is created. Figure 2 shows the static temperature output of the three-axis accelerometer over time during a cold start. A gradual and consistent increase in temperature is observed for all three axes as the system stabilizes. The *x*-axis demonstrates the highest temperature rise, followed by the *z*-axis, while the *y*-axis exhibits the smallest increase. This behavior reflects the varying thermal responses of the axes, which are likely influenced by differences in heat dissipation and material properties. The static conditions of the test ensure that the temperature changes are solely caused by the intrinsic thermal characteristics of the accelerometer, without interference from external accelerations. Due to the influence of temperature variations on the error characteristics of the accelerometer, the errors on the *y*-axis and *z*-axis may increase with temperature fluctuations. The experimental data indicate that, without temperature compensation, an unstable trend under varying temperature conditions is exhibited by the accelerometer output. To address the significant variations in the bias of accelerometer and scale factor caused by temperature effects, an improved polynomial temperature compensation model is proposed. The proposed PCF model incorporates the temperature rate of change to accurately capture the dynamic impact of temperature variations on the accelerometer output. By incorporating the temperature rate of change, the dynamic impact of temperature variations on the accelerometer output is more accurately captured. Traditional temperature compensation methods may require a considerable amount of time to adapt to temperature changes. However, in the PCF model, the compensation parameters are adjusted more sensitively by considering the temperature rate of change. This enables the system to reach a stable state more quickly after startup, reducing the preparation time for SINS and facilitating a rapid transition into operational mode.

The temperature characteristics of QFA are primarily manifested in how environmental temperature variations affect its measurement accuracy, leading to bias drift and scale factor changes. As the temperature changes, the thermal expansion of quartz materials, variations in elastic modulus, and the temperature sensitivity of internal electronic components cause the bias and scale factors to drift and fluctuate, thereby affecting the accuracy of the measurement results. Moreover, the temperature characteristics of QFA may exhibit nonlinear behavior, which becomes more pronounced over a wider temperature range. These temperature effects can be further intensified in rapidly changing thermal environments, leading to delays or drifts in dynamic response. The impact of temperature on the accelerometer can be represented by the current temperature value and temperature-related parameters. The simplified form of the static mathematical model for QFA is as follows:(4)$$A=\frac{E}{K}=B+a_{i}$$
where *A* represents the acceleration value recorded by the instrument, *E* is the accelerometer’s output, *K* denotes the scale factor, *B* refers to the accelerometer’s offset value, and aᵢ represents the input acceleration. The mathematical relationships governing the offset (*B*), scale factor (*K*), and temperature (*T*) of the accelerometer are provided below(5)$$\begin{array}{l} B(T)=S_{B0}+S_{B1}T\\ K(T)=S_{K0}+S_{K1}T \end{array}$$
where, *S*_B0_ represents the zero-order temperature coefficient for the accelerometer bias, while *S*_B1_ denotes the first-order temperature coefficient for the accelerometer bias. Similarly, *S*_K0_ corresponds to the zero-order temperature coefficient for the accelerometer scale factor, and *S*_k1_ represents the first-order temperature coefficient for the accelerometer scale factor.

By substituting (5) into (4), the accelerometer output at different *T* can be obtained as *E*(*T*) as follows:(6)$$\frac{E(T)}{S_{K0}+S_{K1}T}=S_{B0}+S_{B1}T+a_{i}$$

The static output model of QFA can be obtained as follows:(7)$$\frac{E(T)}{K(T)}=B(T)+a_{i}\Leftrightarrow \frac{E}{K}=B+a_{i}$$

The temperature compensation formula can be obtained by combining (6) and (7) as follows:(8)$$a_{i}=\frac{E(T)}{S_{K0}+S_{K1}T}-(S_{B0}+S_{B1}T)=\frac{E}{K}-B$$

Thus, the temperature compensation formula is simplified as follows:(9)$$\begin{array}{l} E=[\frac{E(T)}{S_{K0}+S_{K1}T}+(B-S_{B0}-S_{B1}T)]\times K=\frac{E(T)}{\frac{S_{K0}}{K}+\frac{S_{K1}}{K}T}+(BK-S_{B0}K)-S_{B1}K\times T\\ \Leftrightarrow E=\frac{1}{a+bT}\times E(T)+c+dT \end{array}$$

In SINS, owing to the comparable effects of temperature fluctuations on the bias and scale factor of inertial sensors, the foundational equation for the accelerometer temperature adjustment model is presented as follows:(10)$$E=\frac{E\left(T\right)}{a+b\cdot T}+\left(c+d\cdot T\right)$$
where *E* represents the compensated output value of the inertial device. *T* denotes the current temperature of the inertial device. *E*(*T*) represents the output value of the device at the temperature. *a* + *b*·*T* represents the scale factor compensation of the inertial device, where *a* represents the zeroth-order coefficient and *b* represents the first-order coefficient. *c* + *d*·*T* is the bias compensation of the inertial device, where *c* represents the zeroth-order coefficient and *d* represents the first-order coefficient

Since the inertial device is not only influenced by the temperature, but also by the rate of change of the temperature, the introduction of the temperature change rate is intended to complement the description of the accelerometer’s performance variations in dynamic temperature environments. The temperature change rate affects the thermal response characteristics and dynamic error behavior of the accelerometer, which in turn further impacts the output signal. This is a dynamic factor that must be specifically considered in the temperature compensation model. Modeling based on both temperature and temperature change rate data can enhance the effectiveness of the compensation model under dynamic temperature conditions. Δ*T* is set as the change value of the temperature of the inertial device in the first *n* minutes (*n* = 1, 2, 3…), and the improved temperature compensation model of the inertial device is established as follows:(11)$$E=\frac{E\left(T\right)}{a+b\cdot T}+\left(c+d\cdot T\right)+e\cdot \Delta T$$
where *e* is the temperature change rate coefficient.

### 3.2. Design of the Improved Genetic Algorithm

#### 3.2.1. Basic Principles of Traditional Genetic Algorithm

The genetic algorithm (GA) is a computational model that simulates natural selection and genetic mechanisms from evolutionary biology to search for optimal solutions [34]. The genetic algorithm is characterized by the ability to determine the global optimum of optimization problems, where the optimization results are independent of the initial conditions. It is a domain-independent algorithm with strong robustness, making it well-suited for solving complex optimization problems [35,36]. The next generation of solutions is generated through processes such as natural selection, crossover, and mutation, whereby solutions with low fitness values are gradually eliminated, and the presence of solutions with high fitness values is increased. The basic principles are described as follows below.

A random initial population is generated, consisting of *N* individuals, with each individual representing a potential solution to the problem. The performance of individuals in the problem space is evaluated using the fitness function, denoted as *F*(*x*).

The selection process is based on fitness evaluation, where suitable individuals are chosen from the population to generate the next generation. However, after several generations, the individuals may experience a decrease in mutual differences and loss of diversity. Therefore, the commonly used “roulette wheel selection method” is employed, where the selection probability is determined by the following:(12)$$P(x)=\frac{F(x)}{\sum_{i}F(i)}$$

The selected individuals are placed into a mating pool for crossover as shown in (13). This process involves replacing and recombining parts of the structures from two individuals of the previous generation to generate new offspring. The selection of the individuals is carried out within the mating pool, and the crossover position is also randomly determined:(13)$$Offspring=Crossover\left(Parent1,Parent2\right)$$

Randomness is introduced by the mutation operation, as certain genes of an individual are modified to generate a new offspring. In this process, certain gene values in individual chromosomes are replaced with other gene values, guided by *P*_m_, leading to the creation of a new individual. The global and local searches of search space can be accomplished by the combination of crossover and mutation operations:(14)$$Offspring=Mutation\left(Parent\right)$$

#### 3.2.2. Design of NUMGA

Traditional algorithms are facing challenges such as slow convergence speed, weak local search capability, high number of control variables, and no definitive termination criteria. In traditional GA, the mutation operation is typically applied to each gene position of every individual with a fixed probability. To some extent, randomness can be introduced by this consistent variation strategy, but may lead to the algorithm becoming trapped in local optima when approaching the global optimum.

To address these issues, a non-uniform mutation strategy named as NUMGA was employed in this study. The core idea is to adjust the mutation probability adaptively during the mutation operation. The degree of adaptive variation is introduced in the process of variation, so individual variation is optimized at different stages of evolution. The accelerometer temperature compensation model, described by a polynomial, involves fitting five parameters. The real-value encoding method is utilized with a chromosome size of 5 and a range of gene values from [−1, 1]. The initial population is generated to maintain diversity, with a population size of 1200. In order to explore extensively the solution space and prevent premature convergence, a crossover rate of 0.9 is set, and the mutation rate ranges from 0.05 to 0.2. Additionally, a non-uniform mutation strategy is introduced based on these settings, as illustrated in Figure 3. Furthermore, the fitness function determines the expected results of the genetic algorithm iterations. The fitness function is designed to calculate the difference between the actual accelerometer output value and the compensated value, aiming to achieve the best fit for the temperature variation model. The fitness function calculation (15) is utilized.(15)$$J=\sum_{i=1}^{n}{[E(T)\text{-}E(C)]}^{2}$$
where *E*(*T*) is the actual output of the accelerometer. E(*C*) represents the compensation results provided by the NUMGA-PCF model. A smaller value of *J* indicates that the compensated data from the model closely approximate the measured data. When the subtraction result approaches zero, it indicates that after compensation, the accelerometer output value becomes less sensitive to temperature variations. During the mutation process, a higher mutation probability is assigned to the initial stages. It is helpful to explore the solution space extensively, and potentially excellent solutions are found. As the algorithm advances, the mutation probability is gradually reduced. Therefore, potential solutions have been found during the iteration process, which increases the convergence. The formula for calculating the mutation rate is designed as follows:(16)$$P_{m}=k_{1}*{\left(1-\frac{Gi}{G}\right)}^{k_{2}}+k_{3}$$
where *G_i_* represents the current iteration count. *G* is the total number of iterations. *k*_1_ is the parameter used to adjust the rate of variation. *k*_2_ is used to control the nonlinear characteristics of the variation rate, and *k*_3_ is used to adjust the offset of the variation rate. By adding the mutation magnitude to the original gene value and combining it with (12), the mutated gene value can be obtained as follows:(17)$$Pm(x)=\frac{F(x)}{\sum_{i}F(i)}$$

If the mutated gene value exceeds the allowed range (defined interval), it is trimmed to the valid range. The above steps are repeated until all genes of all individuals have been mutated. In the gene mutation operation of NUMGA, the adjustment of the mutated gene values is performed to ensure that the mutated values remain within a predefined reasonable range. In this study, the temperature change rate is generally low, and its impact on the accelerometer is gradual, while the change rate of the input effects can be significantly higher. The dynamic range caused by such effects needs to be measured based on the actual response capacity of the accelerometer. Therefore, the mutation adjustment strategy in the NUMGA is not designed to simulate actively imposed acceleration changes but rather focuses on constraining drift values caused by passive factors, like temperature changes. This ensures the stability of the iterative results and maintains the physical significance of the model.

## 4. Experimental Validation

The FSINS4X SINS as the carrier was utilized in this experiment for collecting accelerometer data, the QFA model was GJN-11, the performance specifications are shown in Table 1. To ensure the capture of subtle variations in the accelerometer under different temperature conditions, the output frequency of the inertial accelerometer was set to 200 Hz. The temperature and temperature change rate of the accelerometer were measured by using temperature sensors fixed inside the accelerometer. Based on a polynomial temperature error model, a method for collecting temperature compensation data was designed. This method relies on a temperature chamber to simulate environmental temperature changes through heating, maintaining a constant temperature, and cooling phases. In the temperature curve, ambient temperature difference, temperature fluctuations over long periods of operation, and reliability are considered. The setting of the test temperature is shown in Figure 4. Figure 5 shows the picture of the actual test of the equipment in the temperature chamber. The performance parameters of the temperature tank are shown in Table 2. The temperature of the turntable chamber is set to room temperature. After the QFA reaches thermal equilibrium, the data are saved. After reaching thermal equilibrium at 25 °C, the temperature is decreased to −20 °C, and the accelerometer is stabilized for one hour. Subsequently, the temperature is increased to 40 °C, stabilized for one hour, and finally decreased to 25 °C until the end. This temperature range is covered to include the extreme low and high temperature conditions of the accelerometer in most industrial applications, ensuring the effectiveness and reliability of the compensation model under a variety of typical operating conditions. Additionally, the rapid or gradual temperature variations that the accelerometer may encounter in real environments are simulated, and the temperature change rate is set in a temperature chamber. The temperature change rate in this section is set to 1 °C/min. The *x*-axis of the accelerometer in the SINS is pointed to the sky, which is placed in a holding tank or a better airtight box, and fixed so that it remains stationary. The system power is turned on, and data from the inertial device and temperature sensors in three directions are collected. Similarly, the *y*-axis and *z*-axis are oriented upward, respectively. The data from the inertial device and temperature sensors in the *x*, *y*, and *z* directions are collected, respectively.

The data collected from the inertial devices and temperature sensors during the temperature experiments are preprocessed by using the moving average by averaging the surrounding 200 data points to a new smoothed value due to the frequency of the device output being 200 Hz. The moving average method may introduce data lag and loss of signal details. To ensure that the lag effect remains within a reasonable range and minimizes its impact on the overall performance of the compensation model, different window sizes were tested. A window size of 200 data points was found to achieve a good balance between noise reduction and the retention of critical signal details. Taking the *x*-axis accelerometer as an example, when the *x*-axis points towards the sky, the accelerometer output data and temperature synchronous output data are collected. After smoothing, the corresponding data are obtained. A temperature compensation model is established based on the current inertial device temperature *T*, the output value of the inertial device *E*(*T*) at temperature *T*, and the temperature variation value Δ*T*. The temperature error model (10) is fitted by substituting the *x*-axis *E*(*T*), *T*, and Δ*T* of the accelerometer. So, the temperature error model parameters a, b, c, d, and e can be obtained. Similarly, the accelerometers and temperature sensor output data of the *y*-axis and *z*-axis are processed. The temperature variation rate of the thermal chamber is set to 1 °C/min; however, such a rapid response is not achieved by the internal circuitry and the accelerometer. If the time interval is set too short, environmental noise or transient variations are introduced. Conversely, if the interval is too long, significant short-term temperature changes are overlooked. The temperature variation value Δ*T* in the temperature error model is set as the temperature change value every 150 s:(18)$$\Delta T(t)=\left\{\begin{array}{l} T(t)-T(t-149),t\geq 150\\ T(t)-T_{0},1\leq t\leq 150 \end{array}\right.$$
where *t* represents the data sampling time. Δ*T*(*t*) is the temperature change value every 150 s at time *t*. *T*(*t*) represents the temperature value output by the temperature sensor at time *t*, and *T*_0_ represents the reference temperature value. In this case, *T*_0_ can be taken as the temperature value at the initial moment of the cold start of the inertial measurement unit. Taking the *x*-axis accelerometer as an example, based on the previous step, the output data *E*(*T*) of the *x*-axis accelerometer at different temperatures as well as the corresponding temperature data *T* and temperature change data Δ*T* can be obtained. The compensation value approaches the actual output value closely under variable temperature conditions, resulting in a better compensation effect.

## 5. Results and Discussion

### 5.1. Temperature Error Compensation

Figure 6 depicts a comparison between the actual output values of the three axes of the accelerometer. The compensation values obtained using the traditional GA and NUMGA-PCF are also displayed when the *x*-axis points towards the sky. In the actual output values of the accelerometers, a curved pattern is exhibited due to temperature variations. In Figure 6a, noticeable discontinuities in the GA optimized compensation values can be observed at 6000 s and 14,000 s, causing significant changes in individual genes. It indicates that the algorithm is caught in a local optimum and the ability to search globally is lost. Conversely, the compensation values obtained using NUM-GA-PCF closely align with the actual value curve. The blue box in the Figure 6a emphasizes that when zoomed in, the compensation error can be reduced to below 10 μg, demonstrating better consistency with the actual output values. The accelerometer outputs for the *y*-axis and *z*-axis are shown in Figure 6b,c, respectively. The compensation values, are obtained by using the NUMGA-PCF model, exhibit smoother and more continuous trends without noticeable discontinuities or anomalies. This indicates that the dynamic nature of the system can be better adapted by the improved PCF model, which has more reliable compensation effects for the temperature variations are pro-vided. In Figure 6b, there is reasonable consistency between the traditional GA optimized polynomial model and the actual output values, although a lag effect is observed in compensating for temperature step changes and slower recovery. In Figure 6c, the superiority of the NUMGA-PCF model for the *z*-axis accelerometer is further demonstrated by its better adaptability to the dynamic range of the system. The improvement Improved results in for a smoother change in the compensation value, and a quicker response to acceleration changes can be obtained in the *z*-axis direction. It This is the result of more sensitive modeling of the compensation model to temperature changes and more accurate compensation calculations.

### 5.2. Compensation Error

The extent to which the QFA is affected by temperature changes depends on whether the error output value of the QFA approaches 0. This compensation error value is determined by the difference between the compensated values of the QFA obtained through GA and the NUMGA polynomial model. Namely, it is also the output values of the accelerometer after compensation. When the error value tends to 0, the output of the QFA after compensation is not significantly affected by the temperature change. Closer to zero means that the output of the accelerometer is reduced by the degree of temperature influence. The compensation error values for the *x*, *y*, and *z* axes are displayed in Figure 7a–c, respectively. The blue and red lines in each figure depict the accelerometer output values after being compensated with the GA and NUMGA optimized polynomial models. The actual output values are represented by the black line in each figure. It can be clearly observed from the diagrams that the compensated output values effectively reduce the disparity between the flexible accelerometer output and the temperature changes. The red line is the output value after NUMGA-PCF compensation, which is basically stabilized around 0 after being amplified in the X-direction and does not significantly change with temperature. Significant fluctuations are exhibited by the compensation on the *y*-axis QFA around 500 s, 1500 s, and 2500 s, indicating large temperature anomalies in the error after compensation using the traditional GA algorithm. In comparison, the improved error output demonstrates better consistency with the zero scale.

The compensation error of the *z*-axis situation shown in Figure 7c is more pronounced. The NUMGA-PCF compensation maintains consistent stability, while continuous jumps are exhibited by the compensation effect of the traditional GA model in the figure. The observed phenomenon indicates that in the *z*-axis direction, there is significant instability in the compensation of the QFA output by using the traditional GA. Its lack of adaptation to rapid temperature changes is exemplified, and the algorithm itself is mired in flaws. When the *y*-axis QFA and *z*-axis QFA are pointing upwards, the compensation errors for all three axes are generally consistent with those observed when the *x*-axis QFA is pointing upwards. After being optimized with NUMGA, the errors are effectively controlled, and the stability of zero bias and scale factor errors is maintained. The values of the average and standard deviation for each parameter are presented in Table 3. Compared to the traditional compensation method, the standard deviations of the three axes are reduced by 612.24 μg, 60.82 μg, and 875.82 μg after compensation with NUMGA-PCF, respectively. The improvement leads to a difference in average error values before and after optimization of approximately 1 to 2 orders of magnitude. When the *y*-axis QFA and *z*-axis QFA are pointing upwards, the average values and standard deviations of the accelerometer errors for the three optimized axes are below 10 μg. The consistency and the stability in different axes and spatial directions of NUMGA-PCF compensation are demonstrated.

To track the changes in performance of NUMGA over time and iterations, and to evaluate the convergence and stability, a fitness curve is plotted as shown in Figure 8. When the *x*-axis QFA is pointing upwards, the optimized output values of the accelerometers in the X, Y, and Z directions are observed to decrease. From the magnified view in 8a, it can be observed that in the *z*-axis direction, the fitness curve initially converges slowly due to the introduction of NUMGA, which brings higher individual diversity. The fitness curve gradually stabilizes during the early iterations in the traditional GA, but a secondary decline is observed, indicating the possibility of early convergence to a local optimal solution. As the fitness curve reflects the changing trend of individual fitness within the population, larger values indicate poorer fitness. Therefore, after multiple iterations, the fitness curve tends to reach a stable state. After 60 generations, no significant changes are observed in the fitness curve values, as seen in Figure 8b,c. The values of the GA fitness curve (dashed lines) are all higher than the values of NUMGA-PCF fitness (solid lines), with slight differences in the three directions. But overall, a better performance is exhibited by NUMGA in optimizing the polynomial model compared to the traditional GA. The diversity introduced by the non-uniform variance strategy allows for a comprehensive exploration of the search space. In addition, adaptive variance magnitude is employed and NUMGA is more likely to jump out of the local optimal solution and perform a broader global search. Premature convergence is avoided and robustness in global search is better demonstrated.

### 5.3. Static Test Verification

To validate the compensation effectiveness of the proposed NUMGA for the polynomial temperature compensation model, a cold-start static test was conducted. The equipment was placed on a marble platform, as shown in Figure 9. The system was exposed to ambient temperature, the output of the accelerometer was monitored, and real-time compensation applied by using NUMGA-PCF. When the *x*-axis QFA is pointing upwards, the accelerometer output results for the three axes (*x*-axis, *y*-axis, and *z*-axis) were as shown in Figure 10a–c, respectively. The marble platform did not ensure that the equipment was perfectly level. Tilt and installation errors of the accelerometer cause deviations in the output signal. Additionally, after the equipment was powered on, the operation of internal circuits led to gradual heating of the sensors and circuits, resulting in drift. Therefore, under room-temperature conditions (non-thermally controlled), the combined effects of tilt error and temperature variation result in changes in the output signal of the triaxial accelerometer over time. This reflects the accelerometer’s sensitivity to operational conditions, such as temperature changes. The accelerometer output error of the system at different temperature conditions is reduced by the NUMGA-PCF model (red line). It basically fluctuates above and below the 0 line therefore the stabilization time for the cold start is shortened. During the startup, the stability of the GA compensation errors is relatively low, especially in the *y*-axis direction, with continuous jumps occurring at 4000–5000 s. The compensated mean square error curves are shown in Figure 10d. The mean square error (MSE) of NUMGA compensation is less than 0.1 μg. This indicates that the better utility of the NUMGA algorithm in polynomial temperature compensation model is demonstrated, especially in the *x*-axis direction.

## 6. Conclusions

In this study, an NUMGA-PCF temperature compensation model is proposed to effectively compensate for temperature error in the QFA of SINS. Temperature error under different temperature conditions was reduced by PCF modeling. The measurement accuracy and stability of QFA in complex temperature environments were improved. Compared to GA, NUMGA enhances global search capabilities and reduces the risk of premature convergence by introducing a non-uniform variance strategy. In the scenario where the *x*-axis QFA is pointing upwards, the standard deviation of three QFAs were reduced by 612.24 μg, 60.82 μg, and 875.82 μg, respectively. The average errors of NUMGA were all below 10 μg, with a difference of 1 to 2 orders of magnitude compared to the traditional methods. Additionally, the effectiveness of the NUMGA-PCF temperature compensation model was verified, as it can reduce the stabilization time in cold-start tests. The MSE for the *x*-axis pointing upwards was below 0.1 μg. The limitation of the local optimal solution in complex optimization problems that traditional GA may face has been overcome.

## Figures and Tables

**Figure 1 sensors-25-00653-f001:**
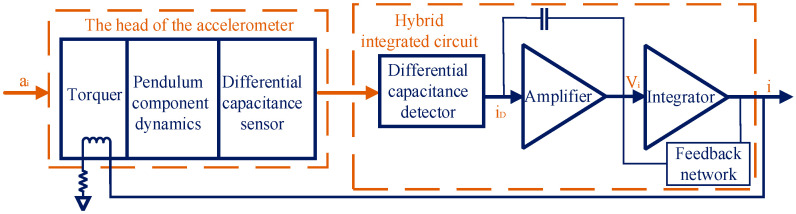
Principle diagram of QFA.

**Figure 2 sensors-25-00653-f002:**
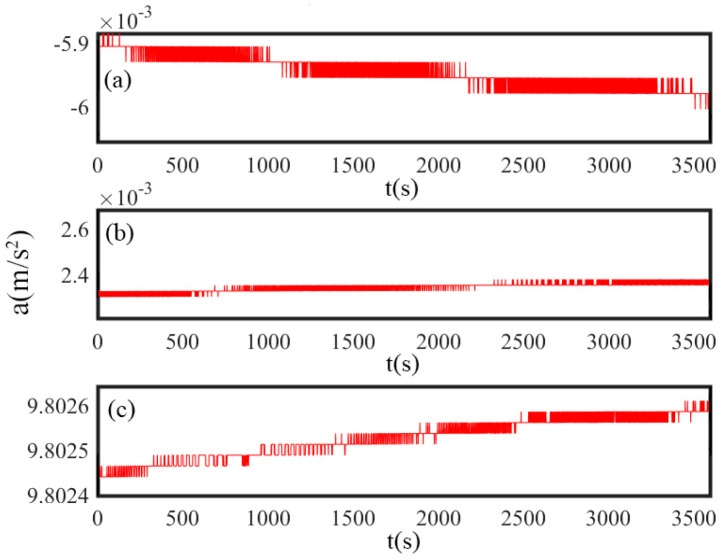
Static Temperature Output of (**a**) *x*-axis accelerometer, (**b**) *y*-axis accelerometer, (**c**) *z*-axis accelerometer.

**Figure 3 sensors-25-00653-f003:**
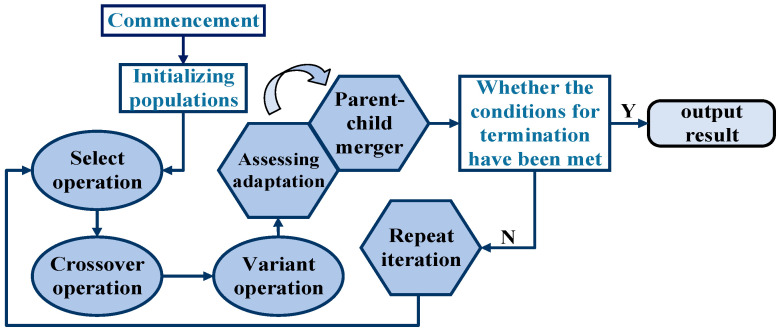
Flowchart of NUMGA.

**Figure 4 sensors-25-00653-f004:**
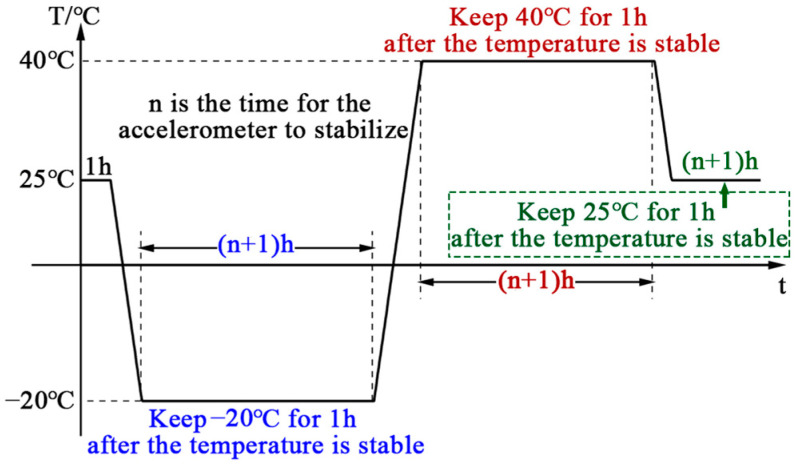
Temperature variation setting diagram.

**Figure 5 sensors-25-00653-f005:**
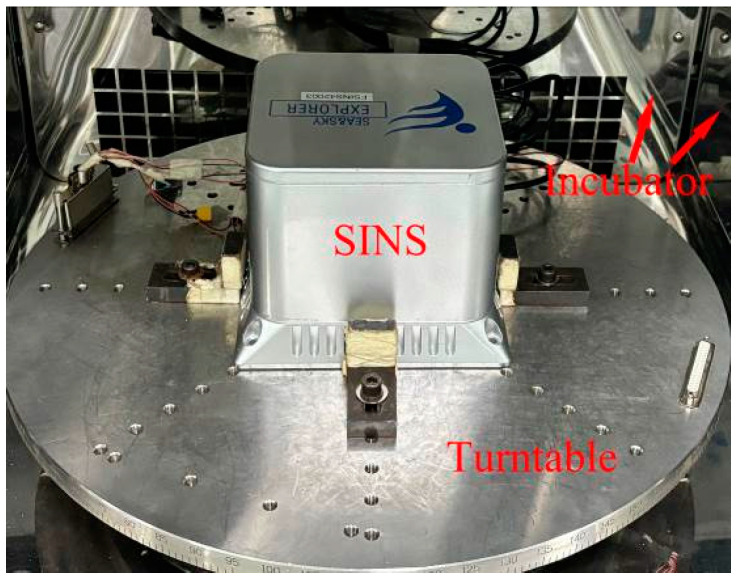
Photo of temperature test chamber test.

**Figure 6 sensors-25-00653-f006:**
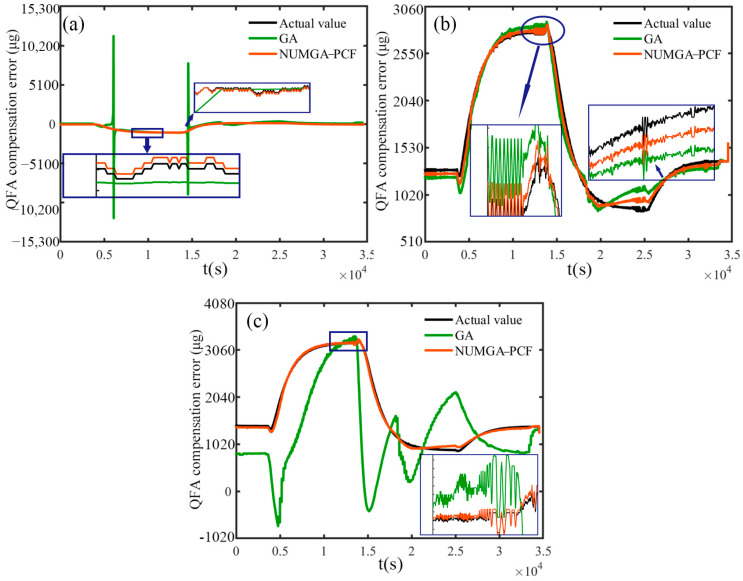
Accelerometer compensation value for (**a**) *x*-axis QFA; (**b**) *y*-axis QFA; (**c**) *z*-axis QFA.

**Figure 7 sensors-25-00653-f007:**
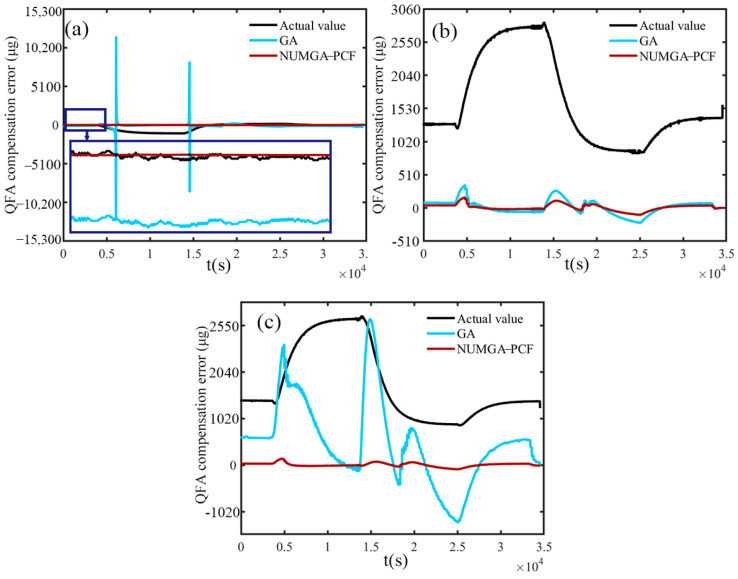
Accelerometer compensation error for (**a**) *x*-axis QFA, (**b**) *y*-axis QFA, (**c**) *z*-axis QFA.

**Figure 8 sensors-25-00653-f008:**
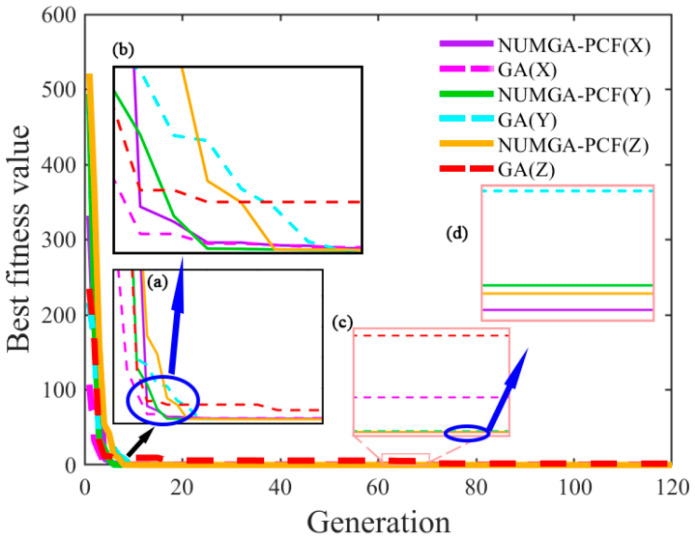
Accelerometer compensation value adaptation curve: (**a**–**d**) are localized magnified images of the fitness curve.

**Figure 9 sensors-25-00653-f009:**
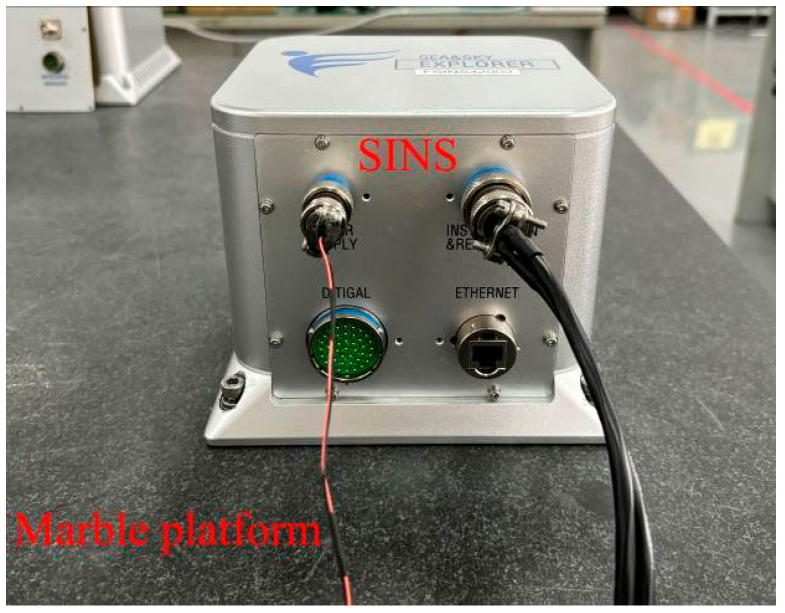
Photo of static cold start test.

**Figure 10 sensors-25-00653-f010:**
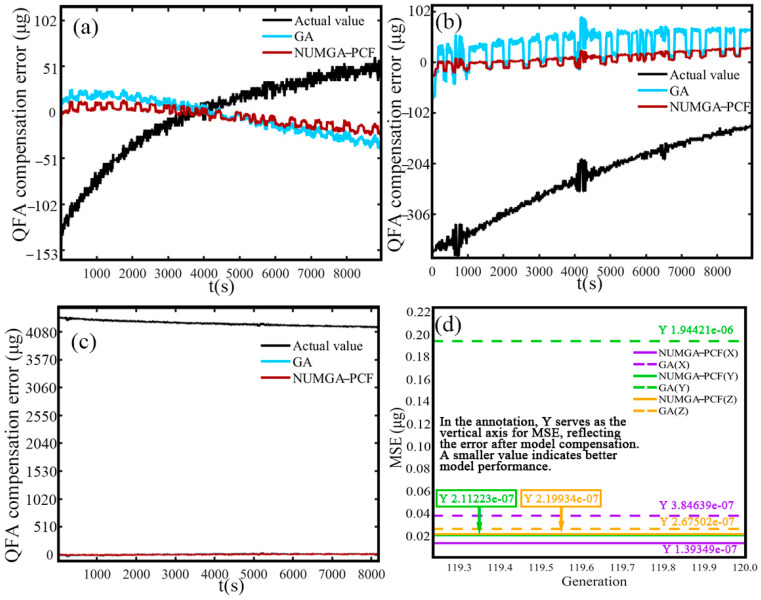
Static test verification results: (**a**) *x*-axis compensation value; (**b**) *y*-axis compensation value; (**c**) *z*-axis compensation value; (**d**) mean square error.

**Table 1 sensors-25-00653-t001:** Main performance index of QFA used in this article.

Model Number	Subproject	Technical Indicator
GJN-11	Range (g)	±30
	Bias (mg)	≤8
	Temperature coefficient (μg/°C)	≤40
	Scale factor (mA/g)Working temperature (°C)	1.0~1.6
		−45~100

**Table 2 sensors-25-00653-t002:** Temperature box performance parameters.

Parameter	Specification	Parameter	Specification
Temperature Range	−55 °C to 100 °C	Temperature Deviation	≤±2 °C
Temperature Control Accuracy	±0.5 °C	Outer Dimensions	W800 × H1390 × D1330 mm
Temperature Uniformity	≤2 °C	Inner Dimensions	W600 × H600 × D600 mm
Heating Rate	Average 3.0 °C/min	Power Supply	AC 380 V 50 HZ
Cooling Rate	Average 3.0 °C/min	Power Consumption	6 KW

**Table 3 sensors-25-00653-t003:** Accelerometer output data.

Experiment	Axial Direction	Primitive Output Value	GA Error Mean Value (μg)	NUMGA-PCF Error Mean Value (μg)	GA Error Standard Deviation (μg)	NUMGA-PCF Standard Deviation (μg)
X pointing upwards	*x*-axis	0.004469	−78.660053	1.903989	620.138035	8.390773
	*y*-axis	0.006690	16.062949	9.912266	109.936154	49.157259
	*z*-axis	0.008218	496.301739	6.398660	916.403852	41.150237
Y pointing upwards	*x*-axis	0.002171	−16.648920	1.438964	73.496613	47.362208
	*y*-axis	0.011598	6.263541	−5.556781	13.236164	11.302934
	*z*-axis	0.008682	22.823716	−14.243499	123.491517	78.839603
Z pointing upwards	*x*-axis	0.001845	−36.249141	23.012903	93.821243	87.621637
	*y*-axis	0.007024	20.266104	10.249569	121.221393	47.356799
	*z*-axis	0.001367	−16.577388	−2.322887	109.981710	7.917484

## Data Availability

The data presented in this study are available on request from the corresponding author.

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
