# Peer review of "Compensation of Temperature-Induced Errors in Quartz Flexible Accelerometers Using a Polynomial-Based Non-Uniform Mutation Genetic Algorithm Framework"

_sensors, 2025, doi:10.3390/s25030653_

Round 1
Reviewer 1 Report
Comments and Suggestions for Authors
This paper introduces an innovative approach to temperature error modeling and compensation for Quartz Flexible Accelerometers (QFA) by proposing a temperature compensation model (NUMGA-PCF) based on a non-uniform mutation genetic algorithm and polynomial curve fitting. Experimental results validate the effectiveness of this compensation model. Overall, the research content is academically valuable, but the paper would benefit from further refinement in the experimental methodology and formatting details.
1. At the end of Section 5.1, there is a repeated sentence expressing the same meaning. It is recommended that redundant content be removed to ensure conciseness.
2. The experimental section could be described in more detail. It is suggested that further information on the experimental design and data collection methods, including the temperature range, temperature change rate, and sampling frequency, be added. For instance, explanations on why the temperature range of -20°C to 40°C was chosen, how the temperature change rate was controlled, and measures taken to ensure data stability would improve the clarity and reproducibility of the experimental design.
3. It is recommended that subsection numbering in Section 5 be checked and corrected to ensure structural clarity and consistency.
4. The paper mentions that the experimental data is smoothed using the moving average method to eliminate high-frequency noise. It is recommended that this process be further described in detail, explaining the reason for using a specific window size (e.g., 200 data points) and its impact on the data smoothing effect. It would also be helpful to discuss whether the moving average method could introduce data lag or loss of detailed signals, which may affect the accuracy of the compensation model.
Author Response
|
Comments 1: At the end of Section 5.1, there is a repeated sentence expressing the same meaning. It is recommended that redundant content be removed to ensure conciseness. |
|
Response 1: At the end of Section 5.1, repeated statements were removed, and other redundant sentences were eliminated. Your careful review in identifying this issue is greatly appreciated, and we are deeply grateful. The modification was marked at the end of Section 5.1, lines 404-407.
|
|
Comments 2: The experimental section could be described in more detail. It is suggested that further information on the experimental design and data collection methods, including the temperature range, temperature change rate, and sampling frequency, be added. For instance, explanations on why the temperature range of -20°C to 40°C was chosen, how the temperature change rate was controlled, and measures taken to ensure data stability would improve the clarity and reproducibility of the experimental design. |
|
Response 2: The temperature range of -20°C to 40°C was chosen to cover the working temperature conditions that the accelerometer may encounter in practical applications. This range represents the extreme low and high temperatures in most industrial applications, ensuring the effectiveness and reliability of the compensation model under various typical working conditions. The modification is marked on page 8 of the paper, lines 331-334. The control of the temperature change rate is implemented to simulate the rapid or gradual temperature variations that the accelerometer may encounter in real environments. The temperature change rate can be set in a temperature chamber, and in this study, it is set to 1°C/min. This part is added to the highlighted green section on page 8, lines 334-337. To ensure the capture of subtle variations in the accelerometer under different temperature conditions, the output frequency of the inertial accelerometer is set to 200 Hz. High-frequency sampling is used to better record the real-time fluctuations in sensor output, thereby more accurately modeling the impact of temperature drift on sensor data. This modification is reflected in the highlighted red section at the end of the second paragraph on page 7, lines 316-317. Comments 3: It is recommended that subsection numbering in Section 5 be checked and corrected to ensure structural clarity and consistency. Response 3: In Section 5, the incorrect numbering of 3.2 and 3.3 appeared. We sincerely apologize for this mistake, and the corrections have been made on pages 10 and 12 where they appeared. Once again, thank you for your careful review. Comments 4: The paper mentions that the experimental data is smoothed using the moving average method to eliminate high-frequency noise. It is recommended that this process be further described in detail, explaining the reason for using a specific window size (e.g., 200 data points) and its impact on the data smoothing effect. It would also be helpful to discuss whether the moving average method could introduce data lag or loss of detailed signals, which may affect the accuracy of the compensation model. Response 4: A window size of 200 data points is chosen for the moving average, primarily based on the sampling frequency of the accelerometer and the characteristics of the noise spectrum. With this selection, most high-frequency noise is effectively smoothed without significantly attenuating the low-frequency components of the signal. A discussion explaining the reasons for this choice and its smoothing effect is added to the end of page 9 (lines 353-357) of the revised manuscript. |

Reviewer 2 Report
Comments and Suggestions for Authors
Thanks to the authors for the interesting material. However, I have some remarks:
1. Lines 159–162: the authors write “Fig. 2 presents the output data of the three-axis temperature sensors when the X-axis is oriented upwards during a cold start. The experimental data indicates that, without temperature compensation, an unstable trend under varying temperature conditions is exhibited of the accelerometer output.” If we compare the values of the output signals along the three axes (Figure 2), their orientation relative to the gravitational acceleration vector g remains unclear. If the X-axis is directed upward, then why is the signal along the other two axes non-zero? Judging by the readings, the Z-axis is directed upward (the readings are close to the value of gravitational acceleration g). It is also necessary to explain why the readings change in steps over time? Perhaps this is due to the nature of the temperature change.
2. Lines 163–165: the authors write “To address the significant variations in the bias of accelerometer and scale factor caused by temperature effects, an improved polynomial temperature compensation model is proposed.” Perhaps, for the convenience of readers, it is worthwhile to provide the characteristics of QFA.
3. Lines 211-212: the authors write "Since the inertial device is not only influenced by the temperature, but also by the rate of change of the temperature." It is necessary to clarify: if the current temperature is already taken into account in expression 10, then why introduce another dependence on its change (which will affect the output signal at a new temperature value)? Or compare the rate of change of temperature and the output signal of the accelerometer.
4. Section 3.2.1 provides a description of the GA algorithm, which is further used to process the output signals of accelerometers. If this algorithm is well known, then it is worth providing references.
5. Lines 291-292: the authors write "If the mutated gene value exceeds the allowed range (defined interval), it is trimmed to the valid range." It is necessary to explain what will happen to the real output signal of the accelerometers if their values change sharply due to the input effect (acceleration), and not temperature? Will it also be cut off? It is worth comparing the rates of change of temperature and input effect for real operating conditions of the accelerometers.
6. Lines 304-305: the authors write "Figure 5 shows the picture of the actual test of the equipment in the temperature chamber." For the convenience of readers, it is worth providing the characteristics of this thermal chamber.
7. Line 306: the authors write "After the fiber optic gyroscope currently reaches thermal equilibrium, data is saved." However, the ms discusses an accelerometer. Perhaps this is an inaccuracy in the presentation and an accelerometer was meant, not a gyroscope.
8. Lines 336-337: the authors write "ΔT(t) is the temperature change value every 150 seconds at time t." It is necessary to explain why this value was chosen?
9. Lines 450-452: the authors write "When the X-axis QFA is pointing upwards, the accelerometer output results for the three axes (X-axis, Y-axis, and Z-axis) are shown in Fig. 10(a), 10(b), and 10(c), respectively." Judging by Fig. 9, the SINS is on a granite slab indoors at room temperature. It is necessary to explain why the output signal (black color) for these three axes changes?
Also, in the future, it is recommended that the authors analyze the reasons for the different nature of the dependencies for the x, y, and z axes of the SINS, since the same type of sensors are located along each of the axes. Perhaps it is worth conducting research first for each sensor separately, and only then assemble the SINS and conduct research on their readings as part of it (repeating the research presented in the ms).
Author Response
|
Comments 1: Lines 159–162: the authors write “Fig. 2 presents the output data of the three-axis temperature sensors when the X-axis is oriented upwards during a cold start. The experimental data indicates that, without temperature compensation, an unstable trend under varying temperature conditions is exhibited of the accelerometer output.” If we compare the values of the output signals along the three axes (Figure 2), their orientation relative to the gravitational acceleration vector g remains unclear. If the X-axis is directed upward, then why is the signal along the other two axes non-zero? Judging by the readings, the Z-axis is directed upward (the readings are close to the value of gravitational acceleration g). It is also necessary to explain why the readings change in steps over time? Perhaps this is due to the nature of the temperature change. |
|
Response 1: The output signal direction of the three-axis accelerometer is determined by its fixed reference coordinate system. In the experiment described in this section, the Z-axis is oriented horizontally upward. Therefore, under static conditions, the gravitational acceleration component primarily acts on the Z-axis, while the outputs of the X-axis and Y-axis, which are subject to errors and drift, are expected to be close to zero but not exactly zero. Due to the influence of temperature variations on the error characteristics of the accelerometer, the errors on the X-axis and Y-axis may increase with temperature fluctuations. This background information has been added to the revised manuscript to further explain the direction of the three-axis signals. This part has been added to the first paragraph of Section 3.1,lines 149-154. |
|
Comments 2: Lines 163–165: the authors write “To address the significant variations in the bias of accelerometer and scale factor caused by temperature effects, an improved polynomial temperature compensation model is proposed.” Perhaps, for the convenience of readers, it is worthwhile to provide the characteristics of QFA. |
|
Response 2: To address the effects of temperature on the bias and scale factor of the accelerometer, the proposed temperature compensation model is based on the analysis of QFA characteristics, aiming to resolve nonlinear deviations in complex thermal environments. In the revised manuscript, a description of the fundamental characteristics of QFA has been added, including how it is affected by temperature and its role in the polynomial compensation model. This part has been added between lines 171-179. Comments 3: Lines 211-212: the authors write "Since the inertial device is not only influenced by the temperature, but also by the rate of change of the temperature." It is necessary to clarify: if the current temperature is already taken into account in expression 10, then why introduce another dependence on its change (which will affect the output signal at a new temperature value)? Or compare the rate of change of temperature and the output signal of the accelerometer. Response 3: The introduction of the temperature change rate is intended to complement the description of the accelerometer's performance variations in dynamic temperature environments. The temperature change rate affects the thermal response characteristics and dynamic error behavior of the accelerometer, which in turn further impacts the output signal. This is a dynamic factor that must be specifically considered in the temperature compensation model. Modeling based on both temperature and temperature change rate data can enhance the effectiveness of the compensation model under dynamic temperature conditions. This part has been added between lines 215-222. Comments 4: Section 3.2.1 provides a description of the GA algorithm, which is further used to process the output signals of accelerometers. If this algorithm is well known, then it is worth providing references. Response 4: References [35]-[37] are added at the end of the article, which mainly explain the use of genetic algorithms to process accelerometer and inertial navigation signals. Comments 5: Lines 291-292: the authors write "If the mutated gene value exceeds the allowed range (defined interval), it is trimmed to the valid range." It is necessary to explain what will happen to the real output signal of the accelerometers if their values change sharply due to the input effect (acceleration), and not temperature? Will it also be cut off? It is worth comparing the rates of change of temperature and input effect for real operating conditions of the accelerometers. Response 5: In the gene mutation operation of the Genetic Algorithm (GA), mutated gene values are adjusted to ensure they remain within a predefined reasonable range. This adjustment mechanism is designed to prevent the generation of discrete values that exceed physical significance or computational constraints during the mutation process. For example, when the operation of an accelerometer is simulated, mutated values are used to represent zero-bias drift caused by temperature changes. These values must be constrained within a reasonable operating range to preserve the model's physical interpretability. Under actual working conditions of the accelerometer, input effects (such as rapid changes in acceleration) and slow drifts caused by environmental temperature changes are two fundamentally different sources of influence. The former involves actively imposed dynamic variations that may cause instantaneous and significant fluctuations in the accelerometer’s output signal, whereas the latter results from gradual changes due to environmental factors. Therefore, in the mutation operation of the GA, the adjustment of mutated values is primarily applied to scenarios involving drifts caused by passive factors like temperature changes, rather than dynamic variations induced by external inputs. The distinction between temperature change rates and input effects (acceleration change rates) is particularly relevant: temperature change rates are typically low, leading to gradual impacts on the accelerometer, while input effects often exhibit significantly higher rates of change, resulting in dynamic ranges that must be assessed based on the accelerometer’s actual response capability. Accordingly, the mutation adjustment strategy in the GA is not designed to simulate actively applied acceleration changes but rather focuses on constraining drift values induced by passive factors like temperature changes. This approach ensures the stability of iterative results and maintains the physical plausibility of the model. The part has been added between lines 302-312 of the paper and has been marked in red. Comments 6: Lines 304-305: the authors write "Figure 5 shows the picture of the actual test of the equipment in the temperature chamber." For the convenience of readers, it is worth providing the characteristics of this thermal chamber. Response 6: The performance parameters of the temperature tank are added in Table 2 on page 9 of the article. Comments 7: Line 306: the authors write "After the fiber optic gyroscope currently reaches thermal equilibrium, data is saved." However, the ms discusses an accelerometer. Perhaps this is an inaccuracy in the presentation and an accelerometer was meant, not a gyroscope. Response 7: Thank you for your thorough review and professional feedback. The error has been corrected to accurately reference the accelerometer instead of the fiber optic gyroscope, lines 326. I have also carefully reviewed the entire text to ensure similar inaccuracies are avoided. Comments 8: Lines 336-337: the authors write "ΔT(t) is the temperature change value every 150 seconds at time t." It is necessary to explain why this value was chosen? Response 8: In this study, a time interval of 150 seconds is selected for calculating the rate of temperature change based on the following considerations: The thermal chamber sets a temperature rate of 1°C/min; however, the internal circuitry and the accelerometer’s temperature response do not achieve such a rapid rate. An appropriate time interval is chosen to better capture the short-term trends in the impact of temperature on accelerometer performance. If a shorter interval, such as one on the order of seconds, is used, environmental noise or transient effects may be introduced, causing instability in the calculated rate of temperature change. Conversely, using a longer interval may result in the neglect of significant short-term temperature variations. By adopting a 150-second interval, the data are smoothed, reducing the impact of noise while still capturing slower temperature variations that may influence sensor performance. This section is added between lines 365-369. Comments 9: Lines 450-452: the authors write "When the X-axis QFA is pointing upwards, the accelerometer output results for the three axes (X-axis, Y-axis, and Z-axis) are shown in Fig. 10(a), 10(b), and 10(c), respectively." Judging by Fig. 9, the SINS is on a granite slab indoors at room temperature. It is necessary to explain why the output signal (black color) for these three axes changes? Response 9: Experiments conducted on a marble platform provide relatively stable physical support but do not guarantee that the equipment is placed in an absolutely horizontal position. This inevitable tilt, combined with installation errors of the accelerometer, causes the output signal to deviate from the ideal values. Additionally, as part of a system, the accelerometer's output is influenced by internal operating conditions. For instance, as the device powers up and operates, heat is generated by the internal circuitry, leading to gradual warming of the sensors and circuits. These temperature changes affect the accelerometer’s scale factor and bias. As a result, even in a room-temperature environment (non-thermally controlled), the triaxial accelerometer's output undergoes changes over time due to the combined effects of tilt errors and temperature variations. It should be noted that these changes are generally slow and have smaller magnitudes compared to the significant variations observed under large temperature gradients in a thermal chamber. Thus, the signal variations under room-temperature conditions reflect both the physical characteristics of the device (e.g., installation errors) and the accelerometer’s sensitivity to operational conditions (e.g., temperature changes). This section is added between lines 484-491. |

Round 2
Reviewer 1 Report
Comments and Suggestions for Authors
The authors have addressed all my concerns. Now it is highly publishable.